# The Interplay between Vascular Function and Sexual Health in Prostate Cancer: The Potential Benefits of Exercise Training

**DOI:** 10.3390/medsci8010011

**Published:** 2020-02-11

**Authors:** Natalie K. Vear, Jeff S. Coombes, Tom G. Bailey, Tina L. Skinner

**Affiliations:** School of Human Movement and Nutrition Sciences, The University of Queensland, Brisbane 4072, Australia; n.vear@uq.edu.au (N.K.V.); jcoombes@uq.edu.au (J.S.C.); tom.bailey@uq.edu.au (T.G.B.)

**Keywords:** physical activity, exercise training, cardiovascular disease, blood pressure, vascular stiffness, endothelial function, sexual function, erectile dysfunction, libido

## Abstract

Prostate cancer and its associated treatments can cause significant and lasting morbidities, such as cardiovascular and sexual dysfunctions. Various interventions have attempted to prevent or mitigate these dysfunctions. This review summarises the available evidence on the effects of exercise training on markers of cardiovascular disease (as assessed via vascular health outcomes) and sexual health in this prevalent cancer population. Current studies predominantly report blood pressure outcomes as a marker of vascular health, as well as various questionnaires assessing sexual health parameters, in men on active treatment (i.e., hormone or radiation therapies) or post-treatment. Preliminary evidence suggests that exercise interventions may elicit improvements in sexual function, but not blood pressure, in these populations. Future studies in more advanced and varied prostate cancer populations (i.e., those on chemotherapies or immunotherapies, or undergoing active surveillance) are required to ascertain the duration, intensity and frequency of exercise that optimises the effects of exercise training on cardiovascular and sexual dysfunctions (and their relationship) in men during and following treatment for prostate cancer.

## 1. Introduction

Prostate cancer and its associated treatments can result in significant and permanent disability as well as increase a man’s risk of all-cause mortality [1]. Men diagnosed with prostate cancer pre-, peri- and post-treatment are at an increased risk of developing highly prevalent complications including forms of cardiovascular disease (CVD) [2,3] and sexual dysfunctions (including erectile dysfunction) [4,5]. Prostate cancer treatments such as androgen derivation therapy (ADT) have been proposed to increase CVD risk due to negative changes in body composition, haemodynamics and blood biomarkers [6,7]. Meanwhile, treatment modalities such as radical prostatectomy and radiation therapies cause sexual dysfunctions in this population due to neural damage and/or detrimental changes to penile haemodynamics [8].

Seventy-to-eighty percent of men with prostate cancer suffer from sexual dysfunction following their cancer diagnosis, inclusive of time on active surveillance [9] and post-treatment [10,11]. Meanwhile, men who undergo ADT are at a 17% increased risk of cardiovascular-related morbidity and mortality following treatment, compared to those not undergoing treatment [12]. In non-cancer populations, CVD risk is a strong determinant of sexual dysfunction [13]. Sexual dysfunction (e.g., erectile dysfunction) is also a predicator of CVD development and risk [14,15]. Further, it has been proposed that declines in vascular health (as assessed by nitric oxide production) are closely linked to declines in sexual functioning in men [16]. A recent meta-analysis in non-cancer populations found that lifestyle interventions aimed at improving CVD risk factors can alleviate erectile dysfunction [17]. Therefore, interventions which are able to effectively prevent and mitigate the development of both sexual and cardiovascular dysfunctions in prostate cancer patients and survivors are warranted.

Differences in anti-cancer treatment types, such as surgery (nerve-sparing vs. non-nerve-sparing), radiation therapy (dose and total surface area) and/or hormone therapy, may also have a significant impact on the treatment response and treatment-related side-effects experienced by men with prostate cancer [18]. Pharmacotherapy interventions can effectively improve cardiovascular health and sexual dysfunctions in cancer populations [19,20]. However, these drugs can be expensive [20], and some patients do not effectively respond to certain pharmacotherapies [19]. As such, alternative non-pharmacological interventions have gained significant attention in recent years, such as those utilising exercise training.

Exercise induces numerous favourable outcomes in prostate cancer patients and survivors [1,21], however limited studies have investigated its effects on cardiovascular health, sexual function or the combination of these outcomes. It has been proposed that changes to early markers of vascular health, such as negative changes to the vascular endothelium, can impair penile haemodynamics and cause sexual dysfunctions such as erectile dysfunction [22]. Endothelial and vascular dysfunction is a precursor to risk of CVD, and a strong predictor of CVD morbidity and mortality [22].

Repetitive increases in blood flow and shear stress induced by exercise training are vital for vascular adaptation. This leads to increases in nitric oxide bioavailability and improves endothelial function [23]. As little as eight weeks of aerobic exercise training have been found to improve cavernous artery nitric oxide bioactivity, and thus improve erectile function [24]. This review therefore presents a summary of the available literature investigating the effects of exercise on CVD risk (as assessed by markers of vascular health) and sexual function in men with prostate cancer.

## 2. Exercise and Vascular Health in Prostate Cancer

High blood pressure has been found to be one of the leading risk factors for global disease burden [25]. In the general population, various forms of structured exercise training have elicited significant improvements in systolic blood pressure (SBP) and diastolic blood pressure (DBP) [26]. However, the relatively limited research that has investigated this in prostate cancer populations has included participants who were either actively on ADT for the duration of the study period [27,28,29,30], or who had completed active treatment (e.g., radiation therapy and ADT) prior to the study initiation [29,31] (Table 1).

Using an observational study design, Uth et al. investigated the effects of five years of habitual unsupervised football training compared to control on blood pressure in men on long-term androgen deprivation therapy (ADT) [39]. No significant between- or within-group differences were observed for either systolic blood pressure (SBP) or diastolic blood pressure (DBP) (all *p* > 0.05) [39].

However, a significantly shorter study by Beydoun et al. [27] found that 10 weeks of supervised exercise training performed at a self-regulated intensity two days per week was effective in significantly decreasing both SBP (−3.4 mmHg, *p* = 0.0044) and DBP (−3.1 mmHg, *p* < 0.0001) in men on ADT [27]. In contrast, a 16-week randomised controlled trial by Culos-Reed et al. [28] found no significant between-group difference for either SBP (*p* = 0.774) or DBP (*p* = 0.833) in men on ADT (>6 months) who completed a low-to-moderate intensity supervised and home-based aerobic and resistance training exercise intervention performed three to five days per week or usual care. However, the exercise group did experience significant within-group improvements in both SBP (−8.9 mmHg, *p* = 0.011) and DBP (−5.6 mmHg, *p* = 0.004) pre- to post-intervention, with the control group also experiencing a significant decrease in DBP (−6.2 mmHg, *p* = 0.007) [28]. The decrease in SBP in the exercise intervention group in this study approached clinical significance, with a 10 mmHg decrease in SBP reported to be associated with a significantly reduced risk of cardiovascular events (RR = 0.80, 95% CI = 0.77–0.83) and all-cause mortality (RR = 0.87, 95% CI = 0.84–0.91) in various clinical populations [40].

These findings are similar to those found in a study by Gaskin et al., which included men who had either completed active treatment less than three months prior to, or were actively on ADT throughout the study period [29]. This study included 12 weeks of supervised moderate intensity aerobic and resistance exercise training two days per week with one day of home-based aerobic exercise training. The control group received verbal advice about general exercise training from their cancer clinician. At 12 weeks, no significant between-group differences were observed for either SBP (−3.8 mmHg, *p* = 0.459) or DBP (−2.0 mmHg, *p* = 0.788). However, significant within-group improvements in both of these blood pressure outcomes were observed in the exercise group, whilst a significant worsening in blood pressure was observed in the control group [29].

Only one study to-date has investigated the effects of a longer-duration exercise intervention on blood pressure in participants who had already completed active treatment (ADT and radiation therapy) prior to study initiation. This study by Galvão and colleagues [31] included supervised (two days/week) and home-based (two days/week) moderate intensity aerobic and resistance exercise training for six months, with completely home-based exercise training performed for 6 months thereafter. The control group received physical activity education material. No significant between-group differences were observed for SBP or DBP at either six (+0.1 mmHg, *p* = 0.822 and +0.2 mmHg, *p* = 0.904, respectively) or 12 months (−1.8 mmHg, *p* = 0.154 and +0.4 mmHg, *p* = 0.726, respectively). Furthermore, SBP and DBP values were shown to be maintained in both groups across all time points [31].

Wall et al. [30] assessed additional markers of vascular health including central aortic wave reflection characteristics and arterial stiffness as measured non-invasively by pulse wave analysis and pulse wave velocity, respectively. Central aortic pulse wave reflection was independently associated with cardiovascular incidence in an ethnically diverse adult population [41]. Meanwhile, central arterial stiffness was found to be a strong predictor of future CVD events, as well as all-cause mortality [42]. Wall and colleagues [30] explored six months of twice-weekly supervised and twice-weekly home-based vigorous intensity aerobic and resistance exercise training compared to usual care in men who had been treated with ADT for greater than two months. No significant between-group differences were observed in SBP (+3.0 mmHg, *p* = 0.508), DBP (−1.0 mmHg, *p* = 0.811), pulse wave analysis outcomes (all *p* > 0.05), or peripheral pulse wave velocity (*p* = 0.0331) at six months. Further, both groups maintained similar CVD risk marker values from baseline to study completion [30].

The currently available literature suggests that low-to-vigorous intensity combined aerobic and resistance exercise training does not currently appear to have a significant benefit on vascular health. Resting blood pressure was not significantly improved following exercise training, when compared to usual care or exercise advice in men with prostate cancer undergoing ADT, or those who had recently completed ADT and radiation therapy. However, significant within-group improvements in blood pressure were observed for low-to-moderate intensity exercise interventions in as short a time as 10 to 16 weeks. This suggests that structured exercise is not detrimental to cardiovascular health during or following treatment for prostate cancer, and may improve vascular health markers in this population. Whether supervised higher-intensity exercise, which has been shown to be superior to moderate intensity continuous training in eliciting improvements in both DBP and SBP in adult populations [43], has similar benefits in men with prostate cancer has yet to be explored.

## 3. Exercise and Sexual Health in Prostate Cancer

A recent observational case–control study by van Stam et al. observed that men who had been treated for prostate cancer reported lower rates of sexual satisfaction compared to age-matched men from the general population [44]. This may be due to the known negative cardio-metabolic side effects of prostate cancer treatments, such as an androgen deprivation therapy [45]. A systematic review by Gerbild et al. suggests that supervised exercise training performed for 40 min four times per week at a moderate-to-vigorous exercise intensity may be effective in improving erectile function in numerous clinical populations [19]. However, of the 10 independent articles summarised within the Gerbild et al. review [19], none included prostate cancer groups. Due to the known neural and haemodynamic complications unique to prostate cancer treatments [8], this specific recommendation may be inappropriate for addressing erectile dysfunction (and other sexual dysfunctions) in prostate cancer groups.

Cross-sectional studies investigating the relationship between habitual physical activity and sexual function in men with prostate cancer have reported mixed findings. Dahn et al. [46] observed that habitual physical activity levels had a significant and positive association with sexual functioning for men undergoing external beam radiation therapy for prostate cancer, but not for men receiving brachytherapy or combination treatments. Meanwhile, Thomas et al. [47] found that following radical radiotherapy, physically active men had a greater ability to have erectile function restored via pharmacotherapeutic intervention. Physically inactive men were more likely to have complete erectile dysfunction, with a blunted response to medication use [47]. Meanwhile, a Canadian study [48] found that meeting the American College of Sports Medicine’s Physical Activity Guidelines for those diagnosed with cancer [49] did not significantly affect the rate of recovery of erectile function in men post-radical prostatectomy [48].

There are a limited number of structured exercise interventional studies that have included assessments of sexual function in prostate cancer populations. The available evidence is limited to men who were either actively on radiation therapy [32,33] or ADT [34] throughout the study period, or who had completed active treatment prior to study initiation [35,36] (Table 1). Sexual function has been assessed in these studies using questionnaires shown to have good clinical validity and reliability [50,51,52], such as the simplified International Index of Erectile Function (IIEF-5), European Organisation for Research and Treatment of Cancer Prostate Cancer-specific Module (EORTC QLQ-PR25) and Expanded Prostate Cancer Index Composite (EPIC-26).

Cormie et al. [34] employed a 12-week twice-weekly supervised moderate-to-vigorous intensity aerobic and resistance exercise training intervention, combined with home-based aerobic exercise training. They observed a significant between-group difference in sexual activity as assessed by the EORTC QLQ-PR25 (+11.7, *p* = 0.045), favouring the exercise group compared to usual care. The exercise group maintained sexual activity scores pre- to post-intervention, whereas control group participants experienced a worsening in sexual activity over the study period [34]. Similar results were observed in a lower-intensity intervention by Ben-Josef et al., which investigated the effects of 12 weeks of twice-weekly low intensity Eischens yoga compared to usual care in men receiving external beam radiation therapy [33]. Significant between-group differences were found in sexual function, as assessed by the IIEF-5, favouring the exercise group at four weeks (*p* = 0.047), but not at 12 weeks (*p* = 0.314) [33]. Similar to Cormie et al. [34], the exercise group also maintained their sexual function whilst the control group worsened over the 12-week period [33]. This maintenance of sexual function is consistent with an earlier paper by Ben-Josef et al. that reported no significant within-group effects of a low intensity Eischens yoga intervention after 12 weeks (*p* = 0.18) [32].

Dieperink et al. used a home-based 20-week pamphlet-based pelvic floor strengthening and resistance training intervention in men who had completed radiation therapy four weeks prior to study randomisation [35]. Compared to usual care, the home-based exercise did not have any significant effect on sexual function, as assessed by the EPIC-26 (*p* = 0.117). In addition, a longer 15-month intervention by Zopf and colleagues [36] was conducted in men who had completed active treatment (surgery, or surgery and radiation therapy) 6 to 12 weeks prior to study enrolment. They compared usual care to once-weekly low-to-moderate intensity supervised aerobic and resistance exercise training, combined with home-based aerobic exercise training for 60 min per week. No significant between-group differences were observed for sexual function for either the EORTC QLQ-PR25 (−5.8, *p* = 0.412) or the IIEF-5 (−2.8, *p* = 0.431) at 15 months. Both the exercise and control groups experienced a significant improvement in sexual function as assessed by the EORTC QLQ-PR25 (+9.8, *p* = 0.008 and +15.6, *p* = 0.008, respectively), but not in the IIEF-5 (both *p* > 0.05). Interestingly, the number of men reported to be sexually active pre- to post-intervention increased from 14 to 16 men in the exercise group, whilst the number decreased from 10 to 8 men in the control group [36].

Collectively, the above articles suggest that exercise interventions can be effective in promoting improvements in sexual function in men with prostate cancer who are undergoing treatment. However, the limited number of studies in men who have completed treatment for prostate cancer suggest that lower-intensity exercise interventions compared to usual care may be insufficient to elicit beneficial changes in sexual health outcomes. The above articles also only included participants with some degree of baseline sexual function, which limits the conclusions that can be drawn from the available literature.

Furthermore, exercise-induced improvements in sexual health have been attributed to improvements in mental health (such as perceived masculinity) in prostate cancer populations [53]. One of the above two articles which reported an improvement in sexual function post-intervention also observed improvements in self-reported quality of life [34]. However, the remaining articles did not observe improvements in quality of life or mental health outcomes [32,33,35,36]. Therefore, mental health is likely to play an important mediating role in exercise-induced changes in sexual health outcomes in men with prostate cancer.

## 4. Exercise and Combined Vascular and Sexual Health in Prostate Cancer

A paucity of studies have investigated the effects of exercise interventions on both markers of vascular function and sexual health in men with prostate cancer (Table 1) [24,37,38]. Cormie et al. [37] assessed the effects of 12 weeks of supervised moderate-to-vigorous intensity aerobic and resistance exercise performed two days per week, combined with home-based aerobic exercise training, compared to usual care in men commencing ADT. A significant between-group difference was observed, favouring the exercise group, for sexual function as assessed by the EORTC QLQ-PR25 (+15.2, *p* = 0.028). On initiating treatment with ADT, the exercise group experienced a non-significant decline (−15.3, *p* = 0.071) in sexual function, whilst the control group experienced a significant decline (−24.3, *p* = 0.012) pre- to post-intervention. No significant between-group differences were observed post-intervention for sexual activity, as assessed by the EORTC QLQ-PR25, or for resting blood pressure (all *p* > 0.05). Sexual activity did decrease significantly in both groups, but to less of an extent in the exercise group (exercise group: −14.5, *p* = 0.015; control group: −20.3, *p* < 0.001). SBP was maintained in both groups during the 12-week study period, with the exercise group also maintaining DBP. However, the control group did experience a significant increase in DBP (+4.0 mmHg, *p* = 0.003) pre- to post-intervention [37].

Jones et al. explored an early marker of CVD risk, brachial artery flow-mediated dilation (FMD) [54], in men who had undergone radical prostatectomy four months prior [24]. Compared to usual care, five days per week of moderate-to-vigorous intensity aerobic exercise training for six months significantly improved FMD (+1.4%, *p* = 0.032). However, this same stimulus did not elicit a significant improvement in sexual function, as assessed by the IIEF-5 (+4.0, *p* = 0.406), compared to usual care. Within-group analyses demonstrated that the exercise group experienced a significant improvement in both brachial artery FMD (*p* = 0.003) and sexual function (*p* = 0.002). The control group also experienced a significant improvement in sexual function (*p* = 0.041) pre- to post-intervention, but not in FMD [24]. These findings suggest that sexual function may have recovered in this population following treatment regardless of the exercise intervention. However, the exercise group also experienced a significant improvement in FMD, which may represent improvements in CVD risk, whilst the usual care group did not [24].

An 11-week group-based walking (once per week) study by Pernar et al. [38] did not lead to significant between-group improvements in SBP, DBP, or sexual function as assessed by a visual analogue scale with the associated question “Do you have problems with your sex life?”. Interestingly, this study observed a non-significant within-group reduction in sexual function in the walking group (+25.8%) and non-significant within-group improvement in the usual care group (−28.6%). These results may be explained in part by the self-reported level of social support provided to participants by their partners. In the intervention group, 21.4% of participants reported low levels of support pre- and post-intervention. Meanwhile only 6.7% of control group participants reported low levels of support pre- and post-intervention. Perceptions of both partner support and patient self-confidence in performing sexual activities have been shown to significantly affect sexual health in men with prostate cancer [10,55] which may help to explain the finding by Pernar et al. [38].

None of the above studies investigated the relationship between vascular function and sexual health in men with prostate cancer. Thus, exercise studies investigating the relationship between changes in vascular function, and other early markers of CVD risk including central arterial stiffness, with sexual health are required. This will help to determine the effects of exercise on the currently ambiguous physiological relationship between sexual and vascular function in this prevalent cancer population. Furthermore, due to the known far-reaching benefits of exercise training in this population [21], patients and survivors should be encouraged to be physically active.

## 5. Conclusions

Treatments for prostate cancer can cause significant cardiovascular and sexual dysfunctions. This review demonstrates that exercise may have a positive effect on sexual health, but not CVD risk, as assessed via changes in blood pressure and vascular function in men during and following active prostate cancer treatment. These results should be interpreted with caution due to the significant heterogeneity evident in both exercise intervention designs and studied prostate cancer groups within the included articles. There is also a paucity of exercise oncological research including men with prostate cancer undergoing watchful waiting, active surveillance, chemotherapies, or immunotherapies. Therefore, further research which includes these understudied prostate cancer groups is required.

Preliminary evidence suggests that exercise interventions may elicit improvements in sexual function, but not blood pressure outcomes, in these populations. Further research including exercise interventions of differing durations, intensities and frequencies are required to identify the optimal exercise dose to elicit protective and beneficial changes in both cardiovascular risk and sexual health in prostate cancer populations. The complexity of these medical and psychological side effects of prostate cancer treatment also means that highly trained allied health professionals with both education on these conditions as well as practical experience are needed to personalise interventions to obtain the best possible outcomes for men in this population. Additionally, studies which include earlier markers of CVD development, employing imaging modalities such as central arterial stiffness, vascular endothelial function and penile peak systolic velocity should be conducted to further delineate the effects of exercise on CVD risk in this population. The relationship between these earlier CVD markers and sexual function should also be investigated. Additionally, biomarkers of cardiovascular remodelling such as testosterone, microalbuminuria, other inflammatory markers (e.g., high-sensitivity c-reactive protein, interleukins −6 and −1 ß) and endothelial-prothrombotic markers (e.g., von Willebrand factor and tissue type plasminogen activator) have been implicated in the relationship between CVD and sexual function in other populations [56,57]. Studies investigating the effects of exercise interventions on these markers are warranted in this population.

Future research is required to guide the development of specific exercise guidelines to optimise the prevention and mitigation of cardiovascular and sexual dysfunctions in this highly prevalent cancer population. This could lead to significant improvements in both quality of life and longevity in men with prostate cancer and their partners.

## Figures and Tables

**Table 1 medsci-08-00011-t001:** Study characteristics of exercise intervention studies.

Article	Study Type	Prostate Cancer Group	Exercise Intervention	Control/Comparison	Outcomes	
					Between-Group	Within-Group
**Vascular Health (Cardiovascular Risk)**						
Beydoun et al. [27]	Three-arm controlled trial	Patients receiving ADT	AEP-supervised exercise training 2 days/week for 10 weeks followed by home-based exercise training for 6 months (*N* = 379); Home-based exercise training for 6 months (*N* = 255)	Education on low-intensity exercise, diet and psychosocial function (*N* = 208)	Both Outcomes NR	SBP Supervised Ex Group Pre < Post 0–10 weeks Home-based Ex & Education Groups NR DBP Supervised EX Group Pre < Post 0–10 weeks Home-based EX & Education Groups NR
Culos-Reed et al. [28]	Two-arm RCT	Patients receiving ADT for ≥6 months	Supervised (by a certified fitness professional) and home-based aerobic and resistance exercise training 3–5 days/week for 16 weeks (*N =* 40)	Usual care (*N =* 22)	SBP EX = CON @ 16 weeks DBP EX = CON @ 16 weeks	SBP EX Pre < Post 0–16 weeks CON Pre = Post 0–16 weeks DBP EX Pre < Post 0–16 weeks CON Pre < Post 0–16 weeks
Galvão et al. [31]	Two-arm RCT	Patients previously treated with ADT and radiation therapy	AEP-supervised aerobic and resistance exercise training 2 days/week and home-based aerobic exercise training 2 days/week for 6 months, followed by home-based aerobic and resistance exercise training for 6 months (*N =* 42 @ 6 month) (*N =* 36 @ 12 month)	Printed physical activity educational material (*N =* 45 @ 6 month) (*N =* 42 @ 12 month)	SBP EX = CON @ 6 months EX = CON @ 12 months DBP EX = CON @ 6 months EX = CON @ 12 months	Both Outcomes NR
Gaskin et al. [29]	Two-arm controlled trial	Completed active treatment 3–12 months prior, or are currently on ADT	AEP-supervised aerobic and resistance exercise training 2 days/week and home-based aerobic and resistance exercise training 1 day/week for 12 weeks (*N =* 53)	Exercise training advice (*N =* 66)	SBP EX = CON @ 12 weeks DBP EX = CON @ 12 weeks	Both Outcomes NR
Wall et al. [30]	Two-arm RCT	Patients receiving ADT for >2 months	AEP-supervised aerobic and resistance exercise training 2 days/week and home-based aerobic and resistance exercise training for 6 months (*N =* 43)	Usual care (*N =* 33)	SBP EX = CON @ 6 months DBP EX = CON @ 6 months Pulse Wave Analysis EX = CON (Central Systolic Pressure, Central Diastolic Pressure, Central Mean Arterial Pressure, Central Augmentation Pressure and Central Augmentation Index) (All *p* > 0.05) @ 6 months Peripheral (carotid-radial) Pulse Wave Velocity EX = CON @ 6 months	All Outcomes NR
Sexual Health						
Ben-Josef et al. [32]	Cohort study	Patients receiving external beam radiation therapy for 6–9 weeks	Eischens yoga sessions 2 days/week for 12 weeks (*N =* 15)	Not Applicable	Not Applicable	Erectile Dysfunction (IIEF-5) Pre = Post 0–12 weeks
Ben-Josef et al. [33]	Two-arm RCT	Patients receiving external beam radiation therapy for 6–9 weeks	Eischens yoga sessions 2 days/week for 12 weeks (*N =* 22)	Usual care (*N =* 28)	Erectile Function (IIEF-5) EX > CON @ 4 weeks EX = CON @ 12 weeks	Erectile Dysfunction (IIEF-5) NR
Cormie et al. [34]	Two-arm RCT	Patients receiving ADT for >2 months	AEP-supervised aerobic and resistance exercise training 2 days/week and home-based aerobic exercise training for 12 weeks (*N =* 29)	Usual care (*N =* 27)	Sexual Activity (EORTC QLQ-PR25) EX > CON @ 12 weeks	Sexual Activity (EORTC QLQ-PR25) NR
Dieperink et al. [35]	Two-arm RCT	Completed radiation therapy (78 Gy in 39 fractions given in five fractions per week) 4 weeks prior to randomisation	Home-based pelvic floor and resistance training 7 days/week for 20 weeks (*N =* 79)	Usual care (*N =* 82)	Sexual Function (EPIC-26) EX = CON @ 20 weeks	Sexual Function (EPIC-26) NR
Zopf et al. [36]	Two-arm controlled trial	Completed active treatment (surgery, or surgery and radiation therapy) 6–12 weeks prior to enrolment	Supervised (by qualified trainers) aerobic and resistance exercise training 1 day/week and home-based exercise 60 min/week for 15 months (*N =* 56)	Usual care (*N =* 29)	Sexual Function (EORTC QLQ-PR25) EX = CON @ 15 months Erectile Function (IIEF-5) EX = CON @ 15 months	Sexual Function (EORTC QLQ-PR25) EX Pre < Post 0–15 months CON Pre < Post 0–15 months Erectile Function (IIEF-5) EX Pre = Post 0–15 months CON Pre = Post 0–15 months Sexually Active EX Pre = 14, Post = 16 CON Pre = 10, Post = 8
Vascular Health (Cardiovascular Risk) and Sexual Health						
Cormie et al. [37]	Two-arm RCT	Patients commencing ADT	Supervised aerobic and resistance exercise training 2 days/week and home-based aerobic exercise training for 12 weeks (*N =* 32)	Usual care (*N =* 31)	Sexual Function (EORTC QLQ-PR25) EX > CON @ 12 weeks Sexual Activity (EORTC QLQ-PR25) EX = CON @ 12 weeks SBP EX = CON @ 12 weeks DBP EX = CON @ 12 weeks	Sexual Function (EORTC QLQ-PR25) EX Pre = Post 0–12 weeks CON Pre > Post 0–12 weeks Sexual Activity (EORTC QLQ-PR25) EX Pre > Post 0–12 weeks CON Pre > Post 0–12 weeks SBP EX Pre = Post 0–12 weeks CON Pre = Post 0–12 weeks DBP EX Pre = Post 0–12 weeks CON Pre > Post 0–12 weeks
Jones et al. [24]	Two-arm RCT	<4 months since radical prostatectomy	Supervised aerobic exercise training 5 days/week for 6 months (*N =* 23)	Usual care (*N =* 23)	BA-FMD EX > CON (Peak % change) @ 6 months EX = CON (Peak % change @ 60 s and Peak % change @ 120 s) @ 6 months Erectile Dysfunction Incidence (IIEF-5 score of < 21) EX = CON @ 6 months Erectile Function (IIEF-5) EX = CON @ 6 months	BA-FMD EX Pre < Post (Peak % change) 0–6 months Pre = Post (Peak % change @ 60 s and Peak % change @ 120 s) 0–6 months CON Pre = Post (Peak % change, Peak % change @ 60 s and Peak % change @ 120 s) 0–6 months Erectile Dysfunction Incidence (IIEF-5) EX Pre < Post 0–6 months CON Pre < Post 0–6 months
Pernar et al. [38]	Two-arm RCT	Completed active treatment >1 month prior	Research nurse-supervised group-based walking sessions 1 day/week and step count goal of 10,000 steps/day for 11 weeks (*N =* 21)	Usual care (*N =* 20)	SBP EX = CON @ 11 weeks DBP EX = CON @ 11 weeks Sexual Function (Visual Analogue Scale-*‘Do you have problems with your sex life?’*) EX = CON @ 11 weeks	All Outcomes NR

ADT: androgen deprivation therapy; AEP: accredited exercise physiologist; BA-FMD: brachial artery flow-mediated dilation; CON: control group; DBP: diastolic blood pressure; EORTC QLQ-PR25: European Organisation for Research and Treatment of Cancer Prostate Cancer-specific Module; EPIC: Expanded Prostate Cancer Index Composite; EX: exercise group; IIEF-5: Simplified International Index of Erectile Function Questionnaire; NR: not reported; RCT; randomised controlled trial; SBP: systolic blood pressure.

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
