# Peer review of "The Interplay between Vascular Function and Sexual Health in Prostate Cancer: The Potential Benefits of Exercise Training"

_medsci, 2020, doi:10.3390/medsci8010011_

Round 1
Reviewer 1 Report
(General): The authorship team are from an established exercise oncology centre in Australia, and have experience researching with several different cancer populations. The review that has been written is appropriate, required, and provides an interesting insight regarding the interplay between vascular function and sexual health that has not yet been explored in the prostate cancer setting, despite each element receiving some attention in the literature in isolation. The authorship team have written a nice review, and I have only minor comments or suggestions for the authors to consider.
(Abstract, Line 22): Recommend adjusting to “… in men on active treatment (i.e. hormone or radiation therapies) or post‐treatment …”.
(Introduction, Line 42): Where it states 70-80% of men with prostate cancer will experience sexual dysfunction; I presume this is total over their lifetime with prostate cancer? I would suggest rewording to: “Seventy‐to‐eighty percent of men with prostate cancer suffer from sexual dysfunction following their cancer diagnosis, inclusive of time on active surveillance [9] and post-treatment [10,11].” – I added “following their cancer diagnosis” / “inclusive of” / and changed “or” to “and”.
(Introduction, Line 43): Appears to be two spaces between “meanwhile” and “men”.
(Introduction): Where it notes that some men do not respond to pharmacotherapies (and so forth), is there any scope for delineating between those who have had nerve sparing surgery or irradiated nerves? Nitric oxide may not help (as with other blood promoting interventions) if the nerves supplying the region are damages due to incision or radiation; thus exercise interventions I presume are more targeted towards those who partially or completely recover from nerve damage; or those who’s nerves were spared? Or both or all? This might just be a worthwhile distinction early, before proceeding to the body of the paper.
(Vascular Function, Line 129-133): Great to see this note. To me it is clear that the reason there is limited improvement to vascular health following these interventions are the clear flaws in the exercise programming (i.e. low to moderate, and often self-managed). As you have correctly highlighted, the intensity of exercise and duration of exercise matters; and your teams’ prior and current work with HIIT would inevitably produce superior outcomes. Certainly worthwhile for future research to investigate.
(Sexual Health): As noted in the comment earlier; the caveat to these exercise and sexual health or sexual function studies is that they included only those with nerve sparring surgery or those with some level of baseline function (i.e. the capacity to improve). There is, as a result, some selection bias in the cohort; I wonder if this can be included in the summary note at the end of this section (Line 191 to 195, roughly)?
(Vascular + Sexual): Nice section that highlights the lack of connectivity in the literature between these two interrelated phenomenon’s with prostate cancer and potential role for exercise medicine; thus an interesting future prospect for future research to pursue.
(Conclusions): Appropriate and justified given the content of the review.
Author Response
(General): The authorship team are from an established exercise oncology centre in Australia, and have experience researching with several different cancer populations. The review that has been written is appropriate, required, and provides an interesting insight regarding the interplay between vascular function and sexual health that has not yet been explored in the prostate cancer setting, despite each element receiving some attention in the literature in isolation. The authorship team have written a nice review, and I have only minor comments or suggestions for the authors to consider.
We thank the reviewer for their kind words.
(Abstract, Line 22): Recommend adjusting to “… in men on active treatment (i.e. hormone or radiation therapies) or post‐treatment …”.
The sentence has been adjusted accordingly.
(Introduction, Line 42): Where it states 70-80% of men with prostate cancer will experience sexual dysfunction; I presume this is total over their lifetime with prostate cancer? I would suggest rewording to: “Seventy‐to‐eighty percent of men with prostate cancer suffer from sexual dysfunction following their cancer diagnosis, inclusive of time on active surveillance [9] and post-treatment [10,11].” – I added “following their cancer diagnosis” / “inclusive of” / and changed “or” to “and”.
This is correct. The sentence has been adjusted accordingly.
(Introduction, Line 43): Appears to be two spaces between “meanwhile” and “men”.
This has been rectified.
(Introduction): Where it notes that some men do not respond to pharmacotherapies (and so forth), is there any scope for delineating between those who have had nerve sparing surgery or irradiated nerves? Nitric oxide may not help (as with other blood promoting interventions) if the nerves supplying the region are damages due to incision or radiation; thus exercise interventions I presume are more targeted towards those who partially or completely recover from nerve damage; or those who’s nerves were spared? Or both or all? This might just be a worthwhile distinction early, before proceeding to the body of the paper.
The authors are not aware of any articles that have investigated differences in pharmacotherapy responses between prostatectomy-only (surgical) and irradiation-only. McCammon et al. (1999) noted that patients having undergone a prostatectomy (n = 203) experienced a greater impact on their sexual relationships than patients having undergone radiotherapy (n = 257). Furthermore, nerve-sparing procedures were found to be more beneficial for erection function than non-nerve-sparing procedures. However, the rates of sexual dysfunction were similar between groups after >12 months. Therefore, the articles included in this review have not placed restrictions on the level of sexual function (regardless of incision or radiation).
The following sentence has been added to the Introduction section of the manuscript (lines 56-59):
“Differences in anti-cancer treatment types, such as surgery (nerve-sparing vs non-nerve-sparing), radiation therapy (dose and total surface area) and/or hormone therapy, may have a significant impact on the treatment response and treatment-related side-effects experienced by men with prostate cancer [18].”
McCammon, K.A, Kolm, P., Main, B., & Schellhammer, P.F. (1999). Comparative quality-of-life analysis after radical prostatectomy or external beam radiation for localized prostate cancer. Urology, 54(3), pp. 509-516
(Vascular Function, Line 129-133): Great to see this note. To me it is clear that the reason there is limited improvement to vascular health following these interventions are the clear flaws in the exercise programming (i.e. low to moderate, and often self-managed). As you have correctly highlighted, the intensity of exercise and duration of exercise matters; and your teams’ prior and current work with HIIT would inevitably produce superior outcomes. Certainly worthwhile for future research to investigate.
Thank you
(Sexual Health): As noted in the comment earlier; the caveat to these exercise and sexual health or sexual function studies is that they included only those with nerve sparring surgery or those with some level of baseline function (i.e. the capacity to improve). There is, as a result, some selection bias in the cohort; I wonder if this can be included in the summary note at the end of this section (Line 191 to 195, roughly)?
We agree with the reviewer. A sentence has been added to the Exercise and Sexual Health in Prostate Cancer section to highlight this limitation. Lines 211-213 now read:
“The above articles only included participants with some degree of baseline sexual function, which limits the conclusions that can be drawn from the available literature.”
(Vascular + Sexual): Nice section that highlights the lack of connectivity in the literature between these two interrelated phenomenon’s with prostate cancer and potential role for exercise medicine; thus an interesting future prospect for future research to pursue.
Thank you
(Conclusions): Appropriate and justified given the content of the review.
Thank you
Reviewer 2 Report
1.
Overall a good paper.
2.
Does exercise improve cardiovascular remodelling and therefore
underlying endothelial dysfunction? If so, shouldnt it take a period
of time to see benefit in men with ED (i.e. not acutely)? Or does
physical activity improve the mental state and therefore this could
translate to a series of biochemical reaction that increases
erectile function in the participants?
3.
Was there any difference between exercise program in men with or
without ADT? And what about those who had prostatectomy vs.
radiation therapy?
4.
Is there any biomarkers of cardiovascular remodelling that improves
performance and sexual activity? e.g. IL, TNF etc? What about
imaging modality e.g. calcium score, angiographic status?
5.
Does men with prostate cancer is at higher risk of ED compared to
normal cardiovascular population? Is this an adverse effect of
cardiometabolic event from ADT?
Author Response
Overall a good paper.
Thank you
Does exercise improve cardiovascular remodelling and therefore underlying endothelial dysfunction? If so, shouldnt it take a period of time to see benefit in men with ED (i.e. not acutely)? Or does physical activity improve the mental state and therefore this could translate to a series of biochemical reaction that increases erectile function in the participants?
Acute exercise elevates blood flow and shear stress which increases nitric oxide availability and endothelial function, and this effect is transient (Dawson et al., 2013). With exercise training, the repetitive increases in blood flow and shear stress are vital for vascular adaptation, including increases in nitric oxide bioavailability and endothelial function (Green et al., 2017). Furthermore, it has been proposed that aerobic exercise training interventions as short as eight weeks in duration can improve nitric oxide bioactivity in cavernous arteries, thereby improving erectile function (Jones et al., 2013).
The following sentences have been added to the Introduction section of the manuscript (lines 70-73) to address this:
“Repetitive increases in blood flow and shear stress induced by exercise training are vital for vascular adaptation. This leads to increases in nitric oxide bioavailability and improves endothelial function [23]. As little as eight weeks of aerobic exercise training have been found to improve cavernous artery nitric oxide bioactivity, and thus improve erectile function [24].”
In men with prostate cancer, it has been proposed that improvements in quality of life, through perceptions of improved masculinity, may play a key role in improving sexual function (Cormie et al., 2013). These positive feelings of masculinity can be facilitated by exercise-induced improvements in factors such as physical appearance, fatigue and peer-support (i.e. exercising with other men) (Cormie et al., 2013). Three articles within this review reported an improvement in sexual function following exercise training, of which two also observed significant improvements in mental health outcomes. Meanwhile, the remaining articles (of which most did not observe a response to exercise training) did not witness any changes in mental health outcomes. Therefore, it is possible that changes in mental state may have influenced sexual function in these men with prostate cancer.
The following sentences have been added to the Exercise and Sexual Health in Prostate Cancer section of the manuscript (lines 214-220) to address this:
“Furthermore, exercise-induced improvements in sexual health have been attributed to improvements in mental health (such as perceived masculinity) in prostate cancer populations [54]. One of the above two articles which reported an improvement in sexual function post-intervention also observed improvements in self-reported quality of life [34]. However, the remaining articles did not observe improvements in quality of life or mental health outcomes [32,33,35,36]. Therefore, mental health is likely to play an important mediating role in exercise-induced changes in sexual health outcomes in men with prostate cancer.”
Cormie, P., Newton, R.U., Taaffe, D.R., Spry, N., & Galvão, D.A. (2013). Exercise therapy for sexual dysfunction after prostate cancer. Nature Reviews Urology, 10(12), 731-736.
Dawson, E., Green, D., Cable, N., & Thijssen, D. (2013). Effects of acute exercise on flow-mediated dilatation in healthy humans. Journal of Applied Physiology (Bethesda, Md. : 1985), 115(11), 1589-1598.
Green, D. J., Hopman, M. T., Padilla, J., Laughlin, M. H., & Thijssen, D. H. (2017). Vascular Adaptation to Exercise in Humans: Role of Hemodynamic Stimuli. Physiological reviews, 97(2), 495–528. doi:10.1152/physrev.00014.2016
Jones, L.W., Hornsby, W.E., Freedland, S.J., Lane, A., West, M.J., Moul, J.W., Ferrandino, M.N., Allen, J.D., Kenjale, A.A., Thomas, S.M., et al. (2014). Effects of nonlinear aerobic training on erectile dysfunction and cardiovascular function following radical prostatectomy for clinically localized prostate cancer. Eur. Urol. 65, 852-855, doi:10.1016/j.eururo.2013.11.009.
Was there any difference between exercise program in men with or without ADT? And what about those who had prostatectomy vs. radiation therapy?
This is an important point of discussion. However, unfortunately insufficient evidence is currently available to determine whether there are any significant differences between prostate cancer treatment groups. This review provides a summary of the available evidence in all exercise oncology articles including prostate cancer populations exploring vascular and/or sexual health outcomes. This lack of available research into specific treatment groups is addressed in lines 273-276 of the Conclusions section as follows:
“There is also a paucity of exercise oncological research including men with prostate cancer undergoing watchful waiting, active surveillance, chemotherapies, or immunotherapies. Therefore, further research which includes these understudied, prostate cancer groups is required.”
Is there any biomarkers of cardiovascular remodelling that improves performance and sexual activity? e.g. IL, TNF etc? What about imaging modality e.g. calcium score, angiographic status?
The reviewer raises an interesting question. There are a number of biomarkers of cardiovascular remodelling, and also imaging modalities, purported to be implicated in cardiovascular disease risk and erectile function. These include testosterone (an independent predictor of cardiovascular disease risk and mortality), microalbuminuria, other inflammatory markers (e.g. high-sensitivity c-reactive protein, interleukins -6 and -1ß) and endothelial-prothrombotic markers (e.g. von Willebrand factor and tissue type plasminogen activator). Furthermore, imaging modalities include penile peak systolic velocity, calcium score and assessments of vascular stiffness via assessments such as pulse wave velocity and carotid intima-media thickness (Vlachopoulos et al., 2006; Vlachopoulos et al., 2015).
The following sentences have been added to the Exercise and Sexual Health in Prostate Cancer section of the manuscript (lines 288-294) to address the biomarkers section of this comment:
“Additionally, biomarkers of cardiovascular remodelling such as testosterone, microalbuminuria, other inflammatory markers (e.g. high-sensitivity c-reactive protein, interleukins -6 and -1ß) and endothelial-prothrombotic markers (e.g. von Willebrand factor and tissue type plasminogen activator) have been implicated in the CVD and sexual function relationship in other populations [57,58]. Studies investigating the effects of exercise interventions on these markers are warranted in this population.”
The following segments have been added in a sentence in the conclusion section of the manuscript (lines 284-287) to address the imaging modalities section of this comment:
“Additionally, studies which include earlier markers of CVD development, employing imaging modalities such as central arterial stiffness, vascular endothelial function and penile peak systolic velocity, should be conducted to further delineate the effects of exercise on CVD risk in this population.”
“employing imaging modalities” and “and penile peak systolic velocity” have been added.
Vlachopoulos, C.; Aznaouridis, K.; Ioakeimidis, N.; Rokkas, K.; Vasiliadou, C.; Alexopoulos, N.; Stefanadi, E.; Askitis, A.; Stefanadis, C. (2006). Unfavourable endothelial and inflammatory state in erectile dysfunction patients with or without coronary artery disease. European heart journal, 27, 2640-2648, doi:10.1093/eurheartj/ehl341.
Vlachopoulos, C., Loakeimidis, N., & Stefanadis, C. (2015). Biomarkers, erectile dysfunction, and cardiovascular risk prediction: The latest of an evolving concept. Asian Journal Of Andrology, 17(1), 17-20.
Does men with prostate cancer is at higher risk of ED compared to normal cardiovascular population? Is this an adverse effect of cardiometabolic event from ADT?
Men who have been treated for prostate cancer reported poorer mental health and sexual health compared to the age-matched general population (Van Stam et al., 2017). The cardio-metabolic side effects of ADT have been linked with various adverse side effects, including sexual dysfunction (Choong & Basaria, 2010).
The following sentences have been added to the Exercise and Sexual Health in Prostate Cancer section of the manuscript (lines 146-149) to address this:
“A recent observational case-control study by van Stam et al. (2017) observed that men who had been treated for prostate cancer reported lower rates of sexual satisfaction compared to age-matched men from the general population [44]. This may be due to the known negative cardio-metabolic side effects of prostate cancer treatments, such as an androgen deprivation therapy [45].”
Choong, K., & Basaria, S. (2010). Emerging cardiometabolic complications of androgen deprivation therapy. The Aging Male, 13(1), 1-9.
Van Stam, M., Van Der Poel, H., Bosch, J., Tillier, C., Horenblas, S., Mols, F., & Aaronson, N. (2017). Prevalence and correlates of mental health problems in prostate cancer survivors: A case-control study comparing survivors with general population peers. Urologic Oncology: Seminars and Original Investigations, 35(8), 531.e1-531.e7.
Round 2
Reviewer 2 Report
Thank you for your revised draft. This has significantly improved the quality of this paper. Well done.